# Interventions to Reduce the Risk of Cardiovascular Disease among Workers: A Systematic Review and Meta-Analysis

**DOI:** 10.3390/ijerph17072267

**Published:** 2020-03-27

**Authors:** Won Ju Hwang, Soo Jin Kang

**Affiliations:** 1College of Nursing Science, Kyung Hee University, 26 Kyunghee-daero, Dongaemun-gu, Seoul 02247, Korea; hwangwj@khu.ac.kr; 2Department of Nursing, Daegu University, 33 Seongdang-ro 50-gil, Nam-gu, Daegu 42400, Korea

**Keywords:** cardiovascular disease, intervention study, meta-analysis, systematic review, workers

## Abstract

This study examined the effect of lifestyle interventions on cardiovascular disease risk factors among workers. The study comprised a systematic review and meta-analysis of controlled trials. Relevant controlled trials were searched, with selections based on the Preferred Reporting Items for Systematic Reviews and Meta-Analyses (PRISMA) guidelines. Risk of bias was assessed using the Scottish Intercollegiate Guidelines Network (SIGN). Of 1174 identified publications, one low-quality study was excluded. Finally, 10 were analyzed. The effect sizes were analyzed for heterogeneity, and random effect models (Hedge’s g) were used. A subgroup analysis was performed on the follow-up point of intervention (≤ 12 months vs. > 12 months). Publication bias was also analyzed. Interventions were effective for systolic (g = 0.66, 95% CI: 0.27-1.60) and diastolic blood pressure (g = 0.63, 95% CI: 0.21–1.06), and BMI (g = 0.71, 95% CI: 0.15-1.11). Interventions were ineffective for weight (g = 0.18, 95% CI: −0.04, 0.40) and LDL-cholesterol (g = 0.46, 95% CI: −0.02, 0.93). There was high heterogeneity between studies (I2 =78.45 to I2 = 94.61). There was no statistically significant publication bias, except for systolic blood pressure. Interventions to reduce risk of cardiovascular disease risk might be effective in improving physical outcomes, but additional high-quality trials are needed in the future.

## 1. Introduction

Cardiovascular diseases (CVD) account for 12.8% of all deaths worldwide and are the leading single cause of death [1]. Mortality due to CVD has been steadily rising over the past 10 years [2]. In Korea, there has also been a consistent increase over the past 10 years. Thus, there is a need to manage CVD more efficiently. In 2015, 166 workers with CVD died, representing a slight decrease from 168 in 2014. Overall, however, the CVD-related relative mortality rate among Korean workers increased from 37.5% in 2014 to 38.4% in 2015 [3]. In addition, CVD among workers is often followed by pneumoconiosis, which was reported to be the second leading cause of morbidity and mortality in 2015. Thus, there is an immediate need for public health efforts to reduce the prevalence of CVD among workers in Korea.

Compared to the general population, workers experience more stress due to heavy workload, lack of exercise, and frequent alcohol consumption; they are also at higher risk for metabolic syndrome due to unhealthy lifestyles [4]. Additionally, they are exposed to work environmental factors such as heavy metals, noise, job stress and physical exertion, which affect the employee in the context of cardiovascular risk [5]. In addition, CVD harms the health of the individual and leads to higher medical costs [6] as well as deterioration of corporate productivity [7]. These factors suggest a need for improved lifestyle management strategies. As a result, various forms of workplace health interventions are required to reduce the risk of CVD among workers. Although some health intervention programs have been implemented for workers, the duration and outcome of programs varies [8]. The factors influencing CVD include both organizational and environmental components, and comprehensive or systematic factors [9]. To prevent and manage CVD among workers, and to provide evidence for planning an effective intervention program, a systematic review and meta-analysis is required. The primary aim of this study is to identify and analyze recent trends in published interventions and to examine the effects of health promotion interventions to reduce the risk of CVD. Thus, the present systematic review and meta-analysis intends to identify and analyze recent trends in interventions and to examine the effects of health promotion interventions in reducing the risk of CVD among workers. In addition, the findings of the present study can be used as a basis for developing and applying intervention programs for workers at risk for CVD.

## 2. Methods 

### 2.1. Search Strategy and Ethical Considerations

The literature search was conducted in December 2018 following the Preferred Reporting Items for Systematic Reviews and Meta-Analyses statement (PRISMA) guidelines [10]. We searched international electronic databases, including PubMed, EMBASE, PsycINFO, the Cumulative Index to Nursing and Allied Health Literature (CINAHL), and the Cochrane Library. Domestic databases included KoreaMed, the Korea medical database (KMbase), and the Korea education and research information service (Riss4u). 

The search was limited to studies published in English and Korean between November 2013 and December 2018. The following keywords and Medical Subject Headings (MeSH) were used: “cardiovascular disease,” “cardiovascular risk,” “cerebro-cardiovascular disease,” “lifestyle intervention,” “risk reduction,” “health behavior,” “health promotion,” “disease prevention,” “health behavior,” “health promotion,” “disease prevention,” “behavioral change,” “exercise,” “physical activity,” “food,” “diet,” “nutrition,” “weight loss,” “stress,” “psychological,” “psychosocial,” “cognitive,” “BMI,” “weight,” “waist,” “cholesterol,” “triglyceride,” “HDL,” “LDL,” “biomarker,” “lipid,” “blood glucose,” “worker,” and “occupation.” The study outcomes selected were blood pressure (systolic and diastolic), body mass index (BMI), waist-to-hip ratio, blood sugar levels, health indicators such as lipid levels, exercise and health behaviors, such as lifestyle practices, and psychological disorders, such as stress and depression. References of the retained studies were manually searched for additional eligible sources. 

As the data used in this systematic review and meta-analysis were obtained from previously published studies, ethical approval and consent were not required.

### 2.2. Inclusion and Exclusion Criteria

The inclusion criteria were as follows: (1) studies about workers; (2) studies that applied interventions to facilitate lifestyle behavioral modifications in diet, physical exercise, and behavior; (3) quasi-experimental or randomized controlled trials (RCTs) with a comparison group; and (4) studies published in English or Korean. Intervention studies that solely focused on pharmacological therapy were excluded. 

### 2.3. Data Extraction 

Relevant data were extracted by two authors (H and K) using a standardized data extraction form developed by the authors. The following data were extracted from the studies: name of the first author, publication year, country, study design, participants, setting, sample size, sample demographics, characteristics of the intervention (number of sessions, duration, length of intervention, and follow-up data points), and study outcomes. 

### 2.4. Quality Assessment 

Two reviewers (H and K) independently assessed the quality of the 11 included studies using the methodology checklist of the Scottish Intercollegiate Guidelines Network (SIGN) [11]. The checklist consists of nine criteria: appropriate research question; randomization, including blinding of treatment allocation; concealment method; similarity between the control group and treatment group; description of the intervention; relevant outcome measurement; dropout rate; intention-to-treat analysis; and confidence of multi-site studies. Based on these criteria, the overall quality of the 11 included studies were classified into three grades: few or none of the criteria were fulfilled, thus the conclusions of the study were likely or very likely to be altered (-); some of the criteria might have been fulfilled, and thus the conclusions were unlikely to be altered (+); and all or most of the criteria had been fulfilled, and the conclusion of the study would definitely not be altered (++) [11].

### 2.5. Data Analysis 

The primary outcomes for the meta-analysis included blood pressure (systolic and diastolic); the secondary outcomes included BMI, weight, and LDL cholesterol. We used a random effects model for meta-analysis to estimate the pooled effect of the intervention. A random effects model was used to calculate the combined effect sizes using Hedges’ *g* equation with the same outcomes, as there was moderate heterogeneity between the studies. This study analyzed the mean difference and standard deviation (SD) of the change scores (i.e., the difference between baseline and end of treatment or follow-up). If the SD of the mean was not reported, it was estimated using 95% confidence intervals (CI) by the principle of estimate method, as previously reported [12], and a correlation coefficient of 0.5 was assumed between the pre-intervention and post-intervention outcome measurements [13]. When multiple groups existed, multiple effect sizes for each intervention-control group were calculated based on the comparisons between the experimental groups and the control group. Heterogeneity was assessed using Q-statistic and I^2^ statistics. The following values were used to determine heterogeneity: I^2^ < 25% (low), I^2^ < 50% (medium), and I^2^ > 75% (high) [13]. The effect of the intervention was examined at the final time-point of the included studies. A subgroup analysis was performed to identify the cause of heterogeneity; therefore, the intervention period was divided into two groups: < 12 months and ≥ 12 months. Data were analyzed using Comprehensive Meta-Analysis software, version 3.1 (Biostat, Inc, Englewood, CO, USA).

A funnel plot was generated to assess publication bias, and Egger’s test of the intercept was performed for further assessment [14]. A *p* value of less than 0.05 was considered statistically significant to indicate evidence of publication bias.

## 3. Results

A total of 1174 publications were retrieved from the electronic searches, and 127 duplicates were removed. Two reviewers independently conducted an initial screening of the titles and abstracts of 990 articles to assess for relevance; 910 articles were excluded because they were irrelevant. Two reviewers independently assessed 80 full-length articles to determine study eligibility. Following this, 69 articles were excluded for the following reasons: no comparison group design and studies that examined pharmacological effects or psychological outcomes. All references were screened by two independent reviewers. Disagreements were resolved through discussion. Finally, 11 studies that met the inclusion criteria were selected for the final review. Figure 1 presents the selection process.

### 3.1. Study Characteristics 

Table 1 provides detailed characteristics of the studies included in this study. Of the 11 included studies, 10 studies were randomized controlled trials (RCTs) and one study was quasi-experimental [15]. The studies were published between 1981 and 2018; five studies were published prior to 2010 [8,15,16,17,18]. Two studies were three-arm trials with two-treatment comparisons [19,20], eight studies were two-arm trials comparing two treatments and a control group, and one study compared an intervention and a wait-list control group [21]. The sample sizes of the studies ranged from 125 to 1292. The locations of studies were the United States (n = 6), Korea (n = 1), Japan (n- = 1), France (n = 1), and the Netherlands (n = 2). The mean age of subjects was similar across studies (39–57 years), except for the study by Cambien et al. [16] in which the mean age was 29. The studies were conducted in community settings [22], workplace settings [8,15,16,17,18,19,21], or at a clinic [20,23,24]. The proportion of male participants ranged from 25% to 100%; one study did not report the gender distribution [16], and two studies included only male participants [17,20]. Study participants included adults with more than one CVD risk factor, including obesity, diabetes, hypertension, type 2 diabetes, etc., except for one [18], which focused on general office workers.

### 3.2. Characteristics of the Interventions

All included studies applied lifestyle intervention programs which included physical exercise and nutritional modifications for reducing the risk of CVD. One study examined a telephone-based intervention [23], one study applied both telephone- and internet-based interventions [19], and one study involved motivational interviewing [24]. The intervention duration (treatment time) ranged from 4 to 12 months. Four studies had an intervention duration of less than six months [19,21,22,24], and seven studies had an intervention duration of more than six months [8,15,16,17,18,20,23]. The follow-up time in eight studies was ≥ 12 months [8,15,16,17,19,20,23,24], and the longest follow-up duration was 24 months [19,20].

Regarding the mode of intervention, the majority of the studies provided individual interventions [8,15,16,18,19,20,24]. Two studies involved both individual and group activities [17,22], and one study provided a group-based intervention [21]. Among the 11 studies, five studies were conducted with remote interventions, using a telephone, the internet, and educational materials [15,16,19,20,23], three studies [8,21,22] were delivered on-worksites, and three studies [17,18,24] combined the two methods. 

### 3.3. Risk of Bias

The quality assessment of the 11 included studies resulted in one study [16] rated as (-), 7 studies [15,17,20,21,22,23,24] rated as (+), and two studies [18,19] rated as (++) (Appendix A, Table A1). The limitations in the methodological quality of each included study included blinding or concealment issues. Non-blinding of subjects to the treatment assignment was a limitation in methodological quality. Only four studies described the blinding of participants or concealment of allocation [18,19,23,24]. Attrition rates for the overall samples based on the final time-point ranged from 0.0% [20] to 74.7% [21]. Of the 11 studies, one low-quality feasibility study [16] was excluded from the meta-analysis due being assessed as at a high risk for sequence generation bias.

### 3.4. Synthesis by Outcomes

Data pooling of all ten studies was undertaken at the final follow-up time-point (Table 2). 

#### 3.4.1. Changes in Systolic Blood Pressure

We conducted a meta-analysis of the 10 studies that measured systolic blood pressure [8,15,17,18,19,20,21,22,23,24]. These studies examined the effectiveness of lifestyle intervention versus standard care for managing systolic blood pressure (Figure 2). Across all included trials, the pooled effect size for systolic blood pressure change at the final time-point was significant (Hedge’s g = 0.66, 95% CI 0.27–1.60, *p* = 0.001), but demonstrated high heterogeneity (I^2^ = 94.2%, *p* < 0.001). 

#### 3.4.2. Changes in Diastolic Blood Pressure

Nine studies [8,15,17,18,19,20,21,22,23,24] measured the change in diastolic blood pressure following the intervention (Figure 2). The pooled effect size for diastolic blood pressure change was 0.63 (95% CI 0.21–1.06, *p* = 0.003), but the heterogeneity was shown to be high (I^2^ = 93.73%, *p* <0.001). 

#### 3.4.3. Changes in BMI 

Six studies [15,17,18,20,21,24] were included in the meta-analysis of BMI changes (Figure 2). There was a significant difference between the lifestyle intervention and control groups (Hedge’s g = 0.71, 95% CI 0.15–1.26, *p* = 0.013), but the heterogeneity was shown to be high (I^2^ = 94.61%, *p* <0.001). 

#### 3.4.4. Changes in Weight

Six studies [8,15,17,19,20,24] were included in the meta-analysis of weight changes. None of the pooled effect sizes were significant at the final time-point. (Hedge’s g = 0.19, 95% CI −0.78–0.46, *p* = 0.166).

#### 3.4.5. Changes in LDL-Cholesterol 

Five studies [8,20,21,23,24] were included in the meta-analysis of LDL cholesterol changes, and no significant change was found (Hedge’s g = 0.46, 95% CI −0.02–0.93, *p* = 0.06).

### 3.5. Publication Bias 

Funnel plot analyses were performed to assess any potential publication bias for systolic blood pressure, diastolic blood pressure, LDL cholesterol, BMI, and weight. The funnel plot is presented in Figure 3. There was a slight asymmetry in the funnel plot for all measured outcomes. Egger’s test results were statistically significant only for systolic blood pressure (*p* = 0.029) (Table 3). This asymmetry was further investigated in the subgroup analysis.

### 3.6. Subgroup Analysis

The intervention follow-up point was divided into two groups (less than 12 months and greater than or equal to 12 months) given the variation in the intervention duration (4 months to 24 months) across studies. In the comparison of studies with interventions for systolic and diastolic blood pressure lasting 12 months (< 12 months) to greater than or equal to 12 months (≥ 12 months), the heterogeneity showed the greatest reduction in I^2^ values for systolic blood pressure, from 96.63% to 2.92% and diastolic blood pressure from 96.12% to 0.00 %. Regarding BMI, there was still moderate heterogeneity, and I^2^ ranged from 97.08% to 45.39%. The results of the subgroup analysis are presented in Table 2. Figure 2 presents the effects of interventions lasting < 12 months on systolic blood pressure. Session duration, length of intervention, intervention provider, and types of intervention were not reported in the sample studies and could therefore not be included in the subgroup analysis. Furthermore, the type of intervention, including individually based (n = 8), group-based (n = 1), remote (n = 5) and on-site (n = 3) interventions, could not be compared in the subgroup analysis due to the lack of studies.

## 4. Discussion

Currently, there are limited high-quality published studies concerning the application and integration of lifestyle interventions in workplace settings. To our knowledge, this is the first meta-analysis to examine the effectiveness of lifestyle interventions on reducing the risk of CVD among workers. While the interventions for workers varied with respect to their contents, methodology, and outcomes, evidence generally suggests that lifestyle interventions, particularly interventions with longer durations, have a beneficial effect on some CVD risk factors. Affected risk factors include systolic blood pressure, diastolic blood pressure, and BMI. Weight and LDL level did not show a significant improvement following lifestyle interventions. These findings are consistent with previous meta-analysis studies [25,26], which reported that lifestyle interventions reduce the risk for CVD among adults with diabetes. 

A thorough search of published studies resulted in only 11 studies of varying quality. Using a quality appraisal instrument, the study by Dekkers et al. [19] was assessed as good quality, while the study by Cambien et al. [16] was assessed as poor quality. Although each study demonstrated the effectiveness of interventions, it is difficult to interpret and compare these findings due to variability between the studies. This variability encompassed the mode of intervention, frequency, intensity, intervention provider, and participants. Because of this, these studies did not provide sufficient intervention information to explore heterogeneity and could not be included in the subgroup analysis. The subgroup analysis, therefore, was conducted concerning only follow-up time of intervention. Moreover, the type of intervention could not be compared in the subgroup analysis due to lack of studies. Therefore, the subgroup analysis included only the length of intervention. 

Studies of interventions lasting less than 12 months showed the largest effect size, but the heterogeneity of this group was high. Compared to interventions lasting less than 12 months, the effect size of long-term intervention (≥ 12 months) was smaller, although it remained positive, and the heterogeneity significantly decreased. This difference between two groups could be because interventions lasting less than 12 months may increase the likelihood of remembering and applying the skills learned to achieve the outcome. Over time, the effects of behavioral interventions were reduced. According to previous studies of interventions for behavioral change, individuals may return to their old behaviors within a year [27,28,29]. Thus, it is important to develop strategies to maintain changes in health behaviors. Further research is needed to examine the critical time-points for intervention and follow-up. 

The lack of consistency in the type of intervention program and the general limitations in the quality of the research reduces the strength of the overall evidence supporting lifestyle interventions for CVD. Although studies are often less robust forms of research, the pooled effects are strong enough to be considered as evidence for health promotion interventions. It appears that there are key elements to developing an integrated intervention program to help workers manage their behavior. 

Education and action learning seem to be the most important components of helping workers respond well to an intervention program. More research needs to be conducted in this area to gain a better understanding. Testing a particular program multiple times would improve the ability to recommend specific, effective, programs. Furthermore, there needs to be more general research into interventions that will help workers manage their health behaviors. Although there are many studies concerning the prevalence and effectiveness of workplace intervention programs, there is a lack of research into this specific area.

We conducted a methodologically rigorous and contemporary search of published studies on the current state of application of lifestyle interventions at worksites; however, there are methodological issues that need to be taken into consideration when interpreting the findings of this review. While the overall attrition rate ranged from 0% [20] to 74.7% [21], an attrition rate greater than 20% was observed in seven studies [15,16,18,19,21,22,24]. In addition, in terms of the risk of bias, the majority of the included studies were assessed as (+) due to lack of blinding or concealment, making response bias likely, which affected the internal validity of the results. 

Lifestyle and integrated interventions, including exercise, nutrition and other components, may reduce the risk of CVD among workers [9,30]. In addition, the development of standardized interventions for objectively monitoring the risk of CVD among workers is identified as a potential benefit of researching lifestyle interventions for workers with known CVD risk factors, including systolic and diastolic blood pressure and BMI [22].

Lastly, the studies in this review were conducted in developed countries and were English publications only. The studies lacked information about the participants’ backgrounds, including education level and work-related characteristics such as blue-collar department or sedentary work; therefore, we cannot conduct subgroup analysis based on employees’ characteristics.

Only 7 of the 11 studies were conducted in workplace settings. Although the participants in the reviewed studies were workers, it was not possible to obtain a sufficient number of relevant articles for this review, despite conducting an extensive systematic electronic search using MeSH terms and keywords. Most of the studies conducted in workplace settings focused on participants’ individual-level characteristics and lacked information about the organizational level; in most cases, such data were not considered or reported. Additionally, “lifestyle interventions” generally refers to active interventions such as allocation of a specific meal type or diet plan, caloric restriction or exercise training. The studies in this analysis are not of these characteristics but are mostly of lifestyle education or counseling programs. According to previous studies [9,31,32], the level of work organization or work environment has a more sustainable effect on the health of workers than individual-based interventions. Future studies should include RCTs that apply organization-level interventions in workplaces.

## 5. Conclusions

There is a paucity of high-quality published research concerning the application of lifestyle interventions for workers to reduce risk of CVD. The present study supports the effectiveness of lifestyle interventions on the risk for CVD among workers at both treatment and follow-up time-points; however, the effects included in this review were based on a small number of RCTs. Future studies need to expand on these findings and continue assessing the effectiveness of interventions for CVD risk among workers.

## Figures and Tables

**Figure 1 ijerph-17-02267-f001:**
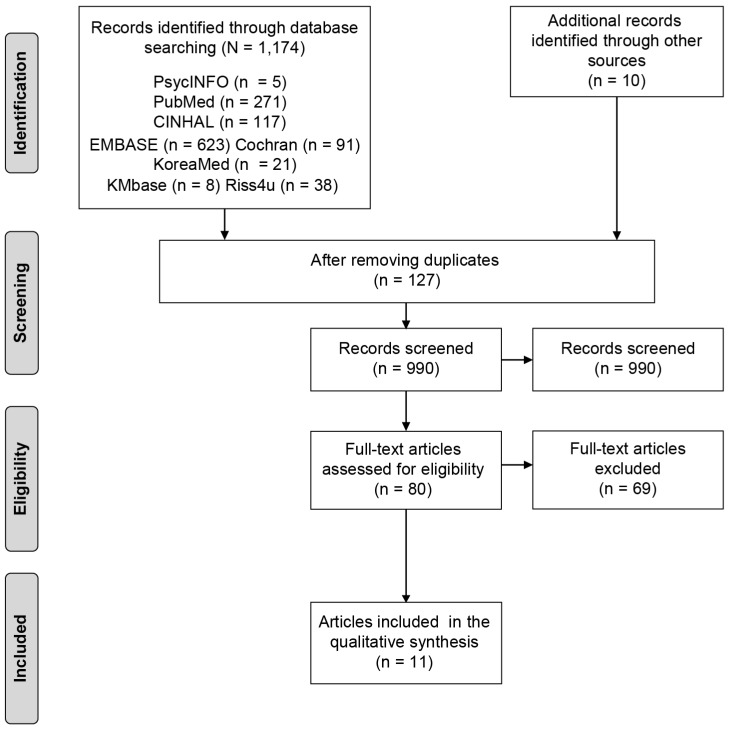
Flow diagram of study selection.

**Figure 2 ijerph-17-02267-f002:**
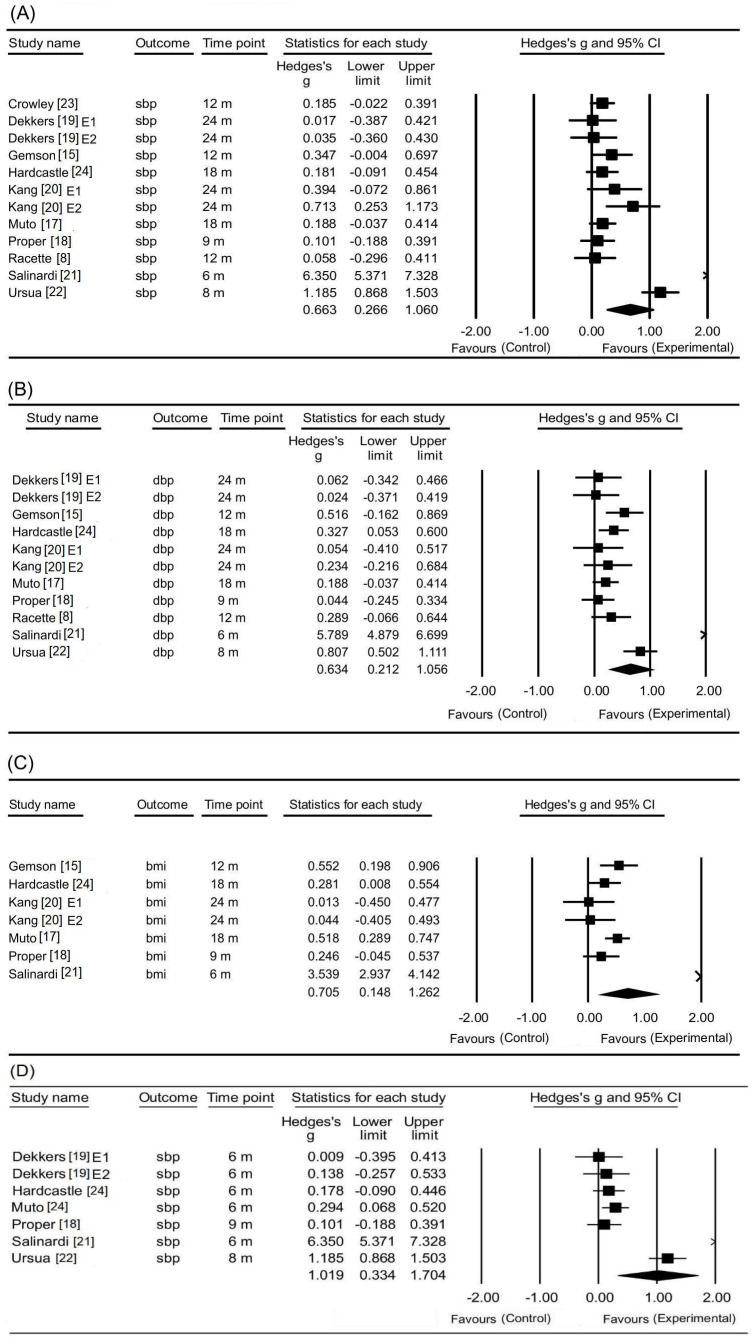
The effect of the intervention. (**A**) Systolic blood pressure at final point; (**B**) Diastolic blood pressure at final point; (**C**) BMI at final point; (**D**)The effect of the intervention on systolic blood pressure < 12 m; E1 = Experimental group 1; E2 = Experimental group 2.

**Figure 3 ijerph-17-02267-f003:**
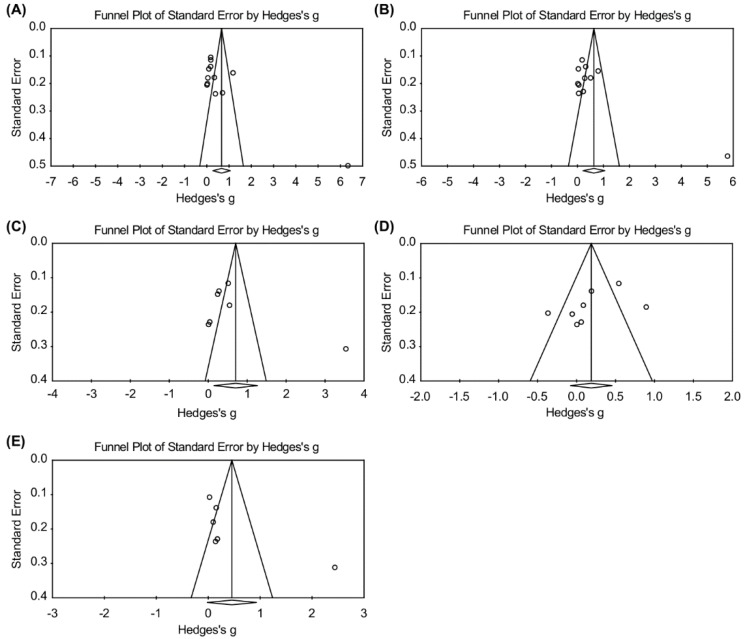
Publication bias at the final point. (**A**) Systolic blood pressure; (**B**) Diastolic blood pressure; (**C**) BMI; (**D**) Weight; (**E**) LDL cholesterol.

**Table 1 ijerph-17-02267-t001:** General Characteristics of the Studies.

1st Author (yr); Ref country	Design	Participants	Settings	No of Participants	Age(yr) Mean ± SD/ Gender = Male (%)	Intervention	Session/ Type	Length Time (minutes)	Duration (Months)/ Follow up Data Points	Outcomes	Quality
Cambein [16] (1981); France	2-arm RCT	Civil servants	Administration department	Exp = 663 Con = 629	Exp 29 ± 3 Con 29 ± 3/ Gender = Not reported	I: Individual lifestyle change program: diet, smoking, physical activity using audio and visual materials C: Not reported	Not reported/ Remote	Not reported	12 months/ Baseline, 12 months, 24 months	Ht, BP(S), Smoking Cholesterol	-
Crowley [23] (2013); USA	2-arm RCT	African Americans with type 2 DM Employees	2 clinics	Exp = 182 Con = 177	Exp 57 ± 12 Con 56 ± 12/ Gender = Male Intervention: 31%, Control:25%	I: Telephone-based nurse-delivered intervention for DM self-management and medication adherence C: Written education materials	Total 12, average 9.9/ Remote	17.1 ± 7.3	12 months/Baseline, 12 months	Primary: BP(S), HbA1c, LDL-C Secondary: medication adherence	+
Dekkers [19] (2011); Netherlands	3-arm RCT	Adults with over-weight (BMI ≥ 25 kg/m^2^) Employees	7 companies (IT, hospital, Insurance, and bank)	Exp1 = 91 Exp2 = 93 Con = 92	Exp1 43 ± 10 Exp2 45 ± 9 Con. 44 ± 9/ Gender = Male Overall: 69.2%	I: Distance-counseling lifestyle intervention by dietician and movement scientists for weight, physical activity and healthy diet Exp1: Phone; Exp2: Internet C: Usual care	Not reported, 10 (max)/ Remote	Not reported	6 months/ Baseline, 6 months, 24 months	BP(S/D), BMI, Wt. Ht, WC, TC, Aerobic fitness level: [VO2 max], Sum of skinfolds	++
Gemson [15] (2008); USA	Quasi-experimental	Hypertension adults Employees (BP ≥ 140 or ≥ 90)	5 sites, 7 financial companies	Exp = 47 (IT, hospital, Insurance, and bank)	Exp 45 ± 9 Con 48 ± 12/ Gender = Male Intervention: 46.8%, Control:51.1%	I: Tailored blood pressure and weight reduction program by nurse based on BP/BMI measuring + pedometer C: Information except physical information	Not available/ Remote	Not available	12 months/ Baseline, 12 months	BP(S/D), BMI, Wt. physical activity, and diet and nutrition behaviors (self-reported)	+
Hardcastle [24] (2013); USA	2 -arm RCT	Adults with CVD risk factors	Primary care center	Exp = 203 Con = 131	Exp 50 ± 1 Con 50 ± 1/ Gender = Not reported	I: Individual motivational intervention by physical activity specialist and dietician for physical activity and nutrition C: Written information	Total 5, average 2/ Remote +on-site	20–30	6 months/ Baseline, 6 months, 18 months	BP(S/D), BMI, Wt, TC, HDL/LDL-C, TG, Physical activity, Nutrition intake	+
Kang [20] (2010); Korea	3 -arm RCT	Male workers with DM risk factors (FPG ≥ 5.6 mmol)	Hospital	Exp1 = 25 Exp2 = 25 Con = 75	Exp1 46 ± 6 Exp2 46 ± 5 Con 47 ± 6/ Gender = Male (100%)	I: Lifestyle intervention for main intervention (3 months) + e-mail nutrition follows- up Exp1: one-year follow-up Exp2: two-year follow-up C: No intervention	5 (counseling)/ Remote	20-30/	12 months/ Baseline, 24 months	BP(S/D), BMI, Wt, TC, HDL/LDL-C, WC, HbA1c, Nutrition intake	+
Muto [17] (2001); Japan	2 -arm RCT	Blue-collar male workers with at least one abnormality in CVD risk factors	Building maintenance company’s worksite branches	Exp = 152 Con = 150	Exp 42 ± 5 Con 43 ± 3/ Gender: Male (100%)	I: Diet and physical activity intervention by health providers (physician, exercise trainer, and coordinators). Individual, group discussion, practice, etc. C: No intervention, written information + annual health checkup	Not available/ Remote +on site	4 days, Every 3 month	12 months/ Baseline, 6, 12 months	BP(S/D), BMI, Wt TC, HDL-C, TG, FB	+
Proper [18] (2003); Netherlands	2-arm RCT	Office employees	3 municipal service	Exp = 131 Con = 168	Exp 44 ± 1 Con 44 ± 1/ Gender = Male Intervention: 74.4%, Control:61.5%	I: Individual counselling on physical activity, nutrition and lifestyle factors; Individual based on PACE protocol C: Written information	7/ Remote +on-site	20	9 months/ Baseline, 9 months	Primary: Energy expenditure, Sport and Leisure-time index, Submaximal HR, Secondary: BMI, TC, BP (S/D), Body fat	++
Racette [8] (2009); USA	2 -arm CohortRCT	Employees with smoke, pre-existing disease (hypertension, diabetes), and medication use	2 Medical center worksites	Exp = 84 Con = 67	Overall 45 ± 9/ Gender = Male Intervention: 23.5%, Control:21.9%	I: Assessment + physical activity and dietary intervention, individual goal setting, group discussion, etc. C: Assessment only	Not reported/ On-site	Not reported	12 months/ Baseline, 6 months, 12 months	BMI, Wt, BP (S/D), TC, HDL/LDL– C, Body fat, glucose	+
Salinardi [21] (2013); USA	2 -arm RCT	Employees with BMI ≥ 25 kg/m^2^	4 worksites Office-based company	Exp = 94 Wait-listed group = 39	Exp 45 ± 1 Con 39 ± 1/ Gender = Male Intervention: 22%, Control:51%	I: Group-based multi-component lifestyle intervention on weight loss and CVD risk factors prevention C: Wait-list weight loss	19/ On-site	60	6 months/ Baseline, 6 months	Primary: Wt Secondary: BP (S/D), BMI BP, TC, TG, glucose HDL/LDL– C	+
Ursua [22] (2018); USA	2 -arm RCT	Filipino Americans with hypertension (BP ≥ 140 or ≥ 90)	A metropolitan	Exp = 112 Con = 128	Exp 54 ± 10 Con 54 ± 10/ Gender = Male Intervention: 39.3%, Control:31.5%	I: CHW led education by Filipino immigrants. Mixed individual and group activities C: Wallet card information	8 (4 group and 4 individual education)/on-site	90	4 months/ Baseline, 4, 8 months	BP (S/D)	+

Note: BMI: body weight index, BP(S): blood pressure (systolic), BP (D): blood pressure (diastolic), C: control, FB: fasting blood, I: intervention, LDL-C: low density lipoprotein cholesterol, HDL-C: high density lipoprotein cholesterol, TC: total cholesterol, Wt: weight.

**Table 2 ijerph-17-02267-t002:** Effect size at the final point, at each point for outcomes.

Outcomes	Final Time Point					T1 (12 months <)					T2 (12 months ≥)				
	Effect Size			Heterogeneity		Effect Size			Heterogeneity		Effect Size			Heterogeneity	
	No. of Studies	Hedges’	95% CI	I2	P	No. of Studies	Hedges’	95% CI	I2	P	No. of Studies	Hedges’	95% CI	I2	P
SBP	10	0.66	0.27, 1.60	94.20	0.000	6	1.02	0.33, 1.70	96.63	0.000	7	0.20	0.10, 0.31	2.92	0.41
DBP	9	0.63	0.21, 1.06	93.73	0.000	6	0.91	0.27, 1.54	96.12	0.000	6	0.23	0.11, 0.35	0.00	0.60
BMI	6	0.71	0.15, 1.26	94.61	0.000	4	1.11	0.22, 2.01	97.08	0.000	4	0.37	0.25, 0.51	45.39	0.120
Weight	6	0.19	−0.78, 0.46	78.45	0.000	4	0.16	−0.29, 0.61	87.80	0.000	6	0.19	−0.08, 0.46	78.45	0.000
LDL	5	0.46	−0.02, 0.93	90.83	0.000	2	1.28	−0.96, 3.52	97.78	0.000	4	0.10	−0.04, 0.23	0.94	0.000

**Table 3 ijerph-17-02267-t003:** Results of publication bias by Egger’s regression test at the final point.

Outcomes	At the Final Point		
	No. of Studies	Egger’s Regression	
		t	*p*-Value
SBP	10	2.54	0.029
DBP	9	2.24	0.052
BMI	6	1.17	0.292
Weight	6	1.52	0.178
LDL	5	1.82	0.143

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
