# Peer review of "Interventions to Reduce the Risk of Cardiovascular Disease among Workers: A Systematic Review and Meta-Analysis"

_ijerph, 2020, doi:10.3390/ijerph17072267_

Round 1
Reviewer 1 Report
The meta-analysis presented for review has been properly performed and described.
It should be published after making a minor adjustment:
In the introduction, the authors pointed out that employees are characterized by an increased cardiovascular risk resulting from behavioral factors, i.e. lifestyle or stress. To avoid simplification, it is also mandatory to mention environmental factors that affect the employee in the context of cardiovascular risk, e.g. heavy metals.
Author Response
Response to Reviewer 2 Comments
Point 1:In the introduction, the authors pointed out that employees are characterized by an increased cardiovascular risk resulting from behavioral factors, i.e. lifestyle or stress. To avoid simplification, it is also mandatory to mention environmental factors that affect the employee in the context of cardiovascular risk, e.g. heavy metals.
Response 1: As you suggested, text edited in Introduction part on page 1. “Also, they are exposed to work environmental factors such as heavy metals, noise, and job stress, physical exertion that affect the employee in the context of cardiovascular risk [5].”
Reviewer 2 Report
This meta-analysis by Hwang et al examined the effect of lifestyle programs on cardiovascular risk factors (blood pressure, BMI, weight, LDL cholesterol) “among workers”. They show that the programs have significant effect on systolic BP, diastolic BP and BMI, but there is high heterogeneity between studies and that there is a need for high-quality trials in the future.
I have the following comments for the authors to address please.
In relation to the terms used in the title as well as in the manuscript text, please review:
- “lifestyle interventions” – this generally refers to active interventions such as allocation of specific meal type/diet plan/caloric restriction and/or exercise training. The studies included in this meta-analysis are not of this nature, are mostly of lifestyle education/counselling programs and should be referred to as such for clarity.
- “Effectiveness to reduce the Risk of CVD” – As the analysis is on the effect on the individual CV risk factors, not on the overall CVD risk nor on the incidence/prevalence of CVD, it should be revised as effect on CV risk factors.
- “workers” – this is very vague, it can be anyone who’s working to earn a living, and I’m not sure what particular groups this term encompasses (?office workers, health workers, labourers, waiters etc etc..) and this is not specified in the details of individual studies. It may be more appropriate to say eg., lifestyle programs offered from workplaces, as most of the analysed studies in this paper seem to have been selected based on this nature.
Could you please explain why you included the study by Crowley 2013 (USA) and Kang 2010 (Korea) which seem to be general DM patients managed in clinic/hospital settings for their DM, that by Hardcastle 2013 (USA) which seem to be general adults with CVD risk factors educated by primary care practice and that by Ursua 2018 (USA) which seem to be general Filipino-Americans with hypertension educated by community health workers (health workers are providers of education). [Table 1]
Could you please specify the type of work of the “workers” listed in the analysed studies (Table 1)?
Please also specify whether the program is delivered on-site or via remote means for all the studies (Table 1). Is there any difference in the effect size between the programs delivered on-site vs remotely?
Please provide P-values in the result sections (3.4.2-3.4.5) and also in the Forrest plots (Figure 2)
The authors described in the text that the program/interventions had significant effects on SBP, DBP, BMI. However, it is depicted in opposite ways in the Forrest plots (Figure 2) (the overall effect favouring controls). Please rectify it correctly.
Please also specify what is “Favours A” and “Favours B” in the figure 2 B and 2 C (Forest plots).
The sub-label for figure 2 D (“D”) is missing.
The study by Cambein (1981) (France) has been listed repeatedly (multiple times) on each page occupied by the Table 1. Please review this.
There are also review the redundant infos in the 2nd rows of the pages (except the first page) of Table 1 .
The paragraphs in Line 120-131 under section 2.5 Data analysis of the manuscript seem to be from author guidelines of the journal!!
Author Response
Revision Response
Thank you for your valuable reviews and comments. We, authors, revised the manuscript as you suggested.
Response to Reviewer 1 Comments
- Point 1: “lifestyle interventions” – this generally refers to active interventions such as allocation of specific meal type/diet plan/caloric restriction and/or exercise training. The studies included in this meta-analysis are not of this nature, are mostly of lifestyle education/counselling programs and should be referred to as such for clarity.Thank you for this point. We revised the sentence on p.1 in the Abstract.
- “This study examined the effect of lifestyle intervention on cardiovascular disease risk factors among workers.”
- Response 1: You have raised an important comment.
- Point 2: “Effectiveness to reduce the Risk of CVD” – As the analysis is on the effect on the individual CV risk factors, not on the overall CVD risk nor on the incidence/prevalence of CVD, it should be revised as effect on CV risk factors.“This study examined the effect of lifestyle intervention on cardiovascular disease risk factors among workers.” Point 3: “workers” – this is very vague, it can be anyone who’s working to earn a living, and I’m not sure what particular groups this term encompasses (?office workers, health workers, labourers, waiters etc etc..) and this is not specified in the details of individual studies. It may be more appropriate to say eg., lifestyle programs offered from workplaces, as most of the analysed studies in this paper seem to have been selected based on this nature.
- “The studies lacked information about the participants’ backgrounds, including education levels and work-related characteristics such as blue-collar department or sedentary workers; therefore, we did not conduct subgroup analysis based on employee characteristics.”
- Response 3: Participants included in this analysis were employees living in communities with a cardiovascular risk factor. However, most studies did not report on employees’ work-related characteristics. For example, Dekker et al.(2011) described adult employees from IT-companies, hospitals, head of insurance, and bank. Kang et al. (2010) recruited from male workers using hospital databases. In table 1, we added work-related information from each study as we can. Also, we added as a limitation on the paragraph in Line. 295-297 on page 5 in Discussion parts.
- Could you please explain why you included the study by Crowley 2013 (USA) and Kang 2010 (Korea) which seem to be general DM patients managed in clinic/hospital settings for their DM, that by Hardcastle 2013 (USA) which seem to be general adults with CVD risk factors educated by primary care practice and that by Ursua 2018 (USA) which seem to be general Filipino-Americans with hypertension educated by community health workers (health workers are providers of education). [Table 1]. Could you please specify the type of work of the “workers” listed in the analysed studies (Table 1)?
- Response 2: Thank you for this point. We revised the sentence on p.1 in Abstract.
Point 4: Please also specify whether the program is delivered on-site or via remote means for all the studies (Table 1). Is there any difference in the effect size between the programs delivered on-site vs remotely?
Response 4: As your comments, we described the parts of the results (3.2 Characteristics of the intervention) on page 4 and Table 1.
We did not conduct the effect size according to the types of intervention. However, we added the reason the parts of the results (3.6 Subgroup analysis)
“Session duration, length of intervention, intervention provider, and types of intervention were not reported in the sample studies and could therefore not be included in the subgroup analysis. Furthermore, the type of intervention, including individually based (n=8) and group-based interventions (n=1), remote (n=5) and on-sites (n=3) could not be compared in the subgroup analysis due to the lack of studies.”
Point 5: Please provide P-values in the result sections (3.4.2-3.4.5) and also in the Forrest plots (Figure 2).
Response 5: We added the p-value in the parts of the results. However, 95 % CI is shown in Figure.
Point 6: The authors described in the text that the program/interventions had significant effects on SBP, DBP, BMI. However, it is depicted in opposite ways in the Forrest plots (Figure 2) (the overall effect favouring controls). Please rectify it correctly.
Response 6: Thank you for your comments. We edited the Figure 2.
Point 7: Please also specify what is “Favours A” and “Favours B” in the figure 2 B and 2 C (Forest plots).
Response 7: We edited the Figure 2.
Point 8: The sub-label for figure 2 D (“D”) is missing.
Response 8: We edited the Figure 2.
Point 9:The study by Cambein (1981) (France) has been listed repeatedly (multiple times) on each page occupied by the Table 1. Please review this.
Response 9: We edited the table 1.
Point 10:There are also review the redundant infos in the 2nd rows of the pages (except the first page) of Table 1.
Response 10: We edited the table 1.
Point 11: The paragraphs in Line 120-131 under section 2.5 Data analysis of the manuscript seem to be from author guidelines of the journal!!
Response 11: We deleted the text